# Sapphire Selective Laser Etching Dependence on Radiation Wavelength and Etchant

**DOI:** 10.3390/mi14010007

**Published:** 2022-12-20

**Authors:** Agnė Butkutė, Romualdas Sirutkaitis, Darius Gailevičius, Domas Paipulas, Valdas Sirutkaitis

**Affiliations:** 1Laser Research Center, Vilnius University, Saulėtekio ave. 10, LT-10223 Vilnius, Lithuania; 2Institute of Biochemistry, Vilnius University, Mokslininkų str. 12, LT-08622 Vilnius, Lithuania

**Keywords:** selective laser etching, 3D laser microfabrication, crystals microprocessing, sapphire 3D structures, femtosecond laser microprocessing

## Abstract

Transparent and high-hardness materials have become the object of wide interest due to their optical and mechanical properties; most notably, concerning technical glasses and crystals. A notable example is sapphire—one of the most rigid materials having impressive mechanical stability, high melting point and a wide transparency window reaching into the UV range, together with impressive laser-induced damage thresholds. Nonetheless, using this material for 3D micro-fabrication is not straightforward due to its brittle nature. On the microscale, selective laser etching (SLE) technology is an appropriate approach for such media. Therefore, we present our research on C-cut crystalline sapphire microprocessing by using femtosecond radiation-induced SLE. Here, we demonstrate a comparison between different wavelength radiation (1030 nm, 515 nm, 343 nm) usage for material modification and various etchants (hydrofluoric acid, sodium hydroxide, potassium hydroxide and sulphuric and phosphoric acid mixture) comparison. Due to the inability to etch crystalline sapphire, regular SLE etchants, such as hydrofluoric acid or potassium hydroxide, have limited adoption in sapphire selective laser etching. Meanwhile, a 78% sulphuric and 22% phosphoric acid mixture at 270 °C temperature is a good alternative for this process. We present the changes in the material after the separate processing steps. After comparing different processing protocols, the perspective is demonstrated for sapphire structure formation.

## 1. Introduction

Sapphire is an attractive material for many applications. It has outstanding mechanical, chemical and optical properties. High-end optical and mechanical components made from sapphire benefit from all of these properties. However, the microprocessing of sapphire is highly complicated. It tends to crack when affected by high-intensity radiation due to the high strain induced during processing. Various laser machining techniques have already been adapted for sapphire micromachining: to make 2D surface structures, dicing [1] or direct laser ablation could be used [2,3]. Direct laser processing can be applied to any material, such as metals [4,5,6,7]. Direct ablation is a relatively efficient and comparably simple process that enables surface functionalization. To improve the quality of laser-processed surfaces, ablation could be combined with additional processes such as laser-induced plasma-assisted ablation [8,9] or laser-induced backside wet etching [10,11,12]. By using ablation in combination with additional thermal processing [13] or direct laser writing in combination with chemical post-processing, optical components such as lenses [14,15] or diffractive optical elements [16,17,18] could be made. Nevertheless, these technologies are limited to the microprocessing and achievement of 2D geometries, and most cannot produce arbitrary-shaped 3D structures. Meanwhile, the selective laser etching (SLE) technique could be a good alternative for 3D structure formation out of sapphire.

SLE is a unique technology that allows the production of 3D structures out of solid-state transparent materials [19]. SLE implementation consists of several steps. First, laser-induced periodic modifications called nanogratings are formed in the bulk of the material by using ultrashort pulsed beams. In amorphous materials, nanaogratings appear as porous modifications, while in crystalline materials, nanaoratings are dominantly regions of amorphized material. Afterwards, laser-modified material is etched out using aggressive etchants such as hydrofluoric acid (HF) or potassium hydroxide (KOH). A similar etching process is applied in the semiconductor industry to make surface structures on silicon [20] or silicon carbide [21]. The capabilities of SLE have been widely investigated on amorphous silica glasses [22,23,24,25]. Critical parameters to describe SLE effectiveness are etching rate and selectivity. The etching rate defines how fast material can be etched, and the selectivity is the ratio between the etching rate of modified and unmodified material. Selectivity describes the highest aspect ratio, which is obtained using a specific SLE processing protocol. Selectivity values above 1000 could be obtained for fused silica by laser parameter optimization [22]. Meanwhile, selectivity values above 2000 can be achieved by introducing a particular burst regime [26] or adding organic solvents to the etchant [27]. SLE is a perfect technology for high aspect ratio structure fabrication. The SLE-made structures could be applied in many areas such as micromechanics [28,29] and microfluidics [30,31,32].

In some works, researchers have already applied SLE to various crystals, such as YAG [33,34,35] or crystalline quartz [36]. More than a decade ago, the first ideas about SLE processed sapphire were published [37]. This publication suggests writing nanogratings inside the sapphire volume and etching it with KOH. A similar procedure was demonstrated with HF acid [38,39]. Etching selectivity values up to 10,000 [40,41,42] have been shown since then. Compared with the value shown in fused silica, this is at least four to five times more. Fused silica features both high etching rates for modified and unmodified regions simultaneously, whereas sapphire features low etching rates for both regions. The significant selectivity in sapphire arises because its etching rate of the unmodified region is negligible, even hard to measure. Even though the selectivity in sapphire is high, only a few simple structures have been shown to come out of sapphire by SLE [39,42,43]. The lack of investigation impedes further development of SLE for transparent materials. In this work, we present research on a sapphire SLE comparison between different approaches such as various laser wavelength and etching protocols. We have chosen widely adopted potassium hydroxide, sodium hydroxide [44], hydrofluoric acid [42], and a mixture of sulphuric and phosphoric acids [43] etching protocols and compared them. Moreover, different radiation wavelengths were tested because the induced nanogratings periodicity is determined by wavelength [45], which later affects the etching properties. Finally, we identified the best protocol for sapphire SLE and made structures using this approach.

## 2. Materials and Methods

SLE experiments were performed with C-cut crystalline sapphire. In these tests, 0.5 mm thickness sapphire substrates were exploited. Many different laser parameters, as well as various etching protocols, were tested. In these experiments, different wavelengths—1st (1030 nm), 2nd (515 nm), and 3rd (343 nm) harmonics of Yb: KGV femtosecond laser (Pharos, Light Conversion Ltd., Lithuania)—were examined. Various focusing optics were chosen to maintain the same focusing conditions and radiation intensity for all tested wavelengths. To focus 1030 nm radiation, a 0.4 NA aspherical lens was used. To focus 515 nm radiation, a 0.2 NA aspherical lens was used, and to focus 343 nm radiatio, an 0.1 NA objective was used. All tested wavelengths were focused to an approximately 1.5 μm beam spot.

Within specific radiation exposition conditions, porous materials modifications called nanogratings can be written inside the volume of glass [46]. Radiation wavelength could change the size of pores and the thickness of the wall between pores. The smaller the wavelength used, the lower the period of the nanogratings [45]. This phenomenon affects the etching properties of the laser-modified volume. In general, the etching mechanism of nanogratings consists of two main aspects. First, the surface area in a material is increased by creating porous structures, therefore increasing reactive surface area, which is one of the reasons it can be etched more efficiently. Another reason is localized amorphization, which makes modified material less resistant to chemical processing [42]. Here, for the disclosed experimental results, circular and linear polarized light was used. It has already been shown that, in fused silica, the light polarization determines the orientation of the nanogratings, which affects the etching rate significantly [46]. Perpendicularly to the scanning direction, polarized light creates nanogratings elongated towards the scanning direction. In this way, prolonged pores enable the penetration and etching of modified material more efficiently than by writing a modification with parallel polarization. A similar tendency of nanogratings’ orientation depending on light polarization has been demonstrated in crystalline sapphire samples, as well [47]. Thus, in single-line experiments, linear polarization perpendicular to the scanning direction was used to maintain a high etching rate. Meanwhile, circularly polarized radiation was used in the experiment where scanning in all XY directions was needed. Sample positioning was performed by using XYZ linear positioning stages (Aerotech, Pittsburgh, PA, USA).

In this work, written nanogratings were etched with various chemicals such as a 35% potassium hydroxide (KOH) mixture with water, a 25% sodium hydroxide (NaOH) mixture with water, and a 48% hydrofluoric acid (HF) and 78% sulphuric, and 22% phosphoric acid mixture (H_2_SO_4_ and H_3_PO_4_). Utilization of all of these etchants differs: KOH and NaOH were used at 90 °C, HF was used at the ambient temperature (20 °C), and H_2_SO_4_ and H_3_PO_4_ were used at 270 °C. The performance of these etchants in the described condition has already been presented widely in other works [42,43,44]. However, etching substances have different properties, notably the boiling point. For water-based alkali mixtures such as KOH or NaOH [48], it is around 120 °C. For HF, it is 106 °C. Meanwhile, the sulphuric acid boiling point is above 300 °C, and the phosphoric acid boiling temperature is above 150 °C. Therefore, different temperatures are required to obtain optimal etching efficiency for these chemicals. Thus, four separate well-known etching protocols were examined and compared for sapphire etching.

First of all, principal experiments of single lines in the nXY plane were performed to test how pulse energy (100, 200, 300, 400 and 500 nJ), radiation wavelength (1030 nm, 515 nm, 343 nm), and the used etchants (NaOH, KOH, HF, H_2_SO_4_ and H_3_PO_4_) affect the etching rate. In this experiment, pulse duration (200 fs) and pulse repetition rate (610 kHz) remain constant. The light was linearly polarized perpendicularly to the scanning direction to obtain etching rates that were as high as possible. The experiment’s main idea was to write single lines inside the bulk of the sapphire. Then, the plate was divided perpendicularly to the written lines in the middle. Finally, the side of the divided plate was polished (using a fine diamond powder on lapping machine), and modifications were etched. The etching rate was evaluated by measuring the length of the etched channel and dividing it by the etching time. This experiment gives us data about the etching rate of the modified material. The idea of this test is shown in Figure 1a. Additionally, during the same experiment, etching rates of unmodified material were evaluated by measuring substrate thickness before and after etching. The etching rate of unaffected material allows us to estimate selectivity in each case. Selectivity is defined as a ratio of modified and unmodified material etching rates. However, since the unmodified material etching rate is negligible in NaOH, KOH, and HF, it cannot be accurately evaluated. For these etching rates, the material was etched for 24 h and the etched material’s thickness was around 1 μm, which is on a scale of fidelity of the used device. Thus, we only can think about possible boundaries of selectivity of each etching protocol.

Subsequently, etching experiments of 3D structures were executed. During this experiment, cylindrical 1 mm diameter 3D structures were written into a plate of sapphire. The scheme of this experiment is depicted in Figure 1b. This experiment tested etching dependency on radiation wavelengths, pulse duration (200 fs-1 ps), and different etchants, to find a protocol for 3D structure fabrication. Finally, when optimal parameters for the etching were found, the morphology of the modified areas was investigated after and before the etching process. For that, 10 μm wide surface modifications were written and etched in the various previously mentioned etchants. The scheme of this experiment is shown in Figure 1c. These morphologies provide some insights into why some etchants are not suitable for sapphire 3D structures’ fabrication. After all, a few 3D structures were fabricated out of sapphire using the same setup and employing the found fabrication protocol.

## 3. Results

The results of single line tests are discussed first. The results of this experiment are shown in Figure 2a. After a single lines investigation, we have found that the highest etching rate (200 μm/h) is obtained with an H_2_SO_4_ and H_3_PO_4_ mixture when the material is irradiated with 1st harmonic radiation and the highest tested pulse energy of 500 nJ, which corresponds to 28.2 J/cm^2^ energy density. Meanwhile, lower etching rates up to 100 μm/h were obtained with NaOH and KOH enchant. The lowest etching rates of up to 50 μm/h were demonstrated by modifications etched with HF. In all the cases, the modification written with 1st harmonic radiation showed the highest etching rates, which was at least a few percent higher than those of the results achieved with the 2nd harmonic, and at least a few times higher than the etching rates obtained with the 3rd harmonic. This result could be explained by the higher periodicity of the nanogratings. The nanogratings’ period is equal to λ/2n, where λ is the wavelength of used radiation and *n*- is the refractive index of the material [49]. Nanogratings are the result of the interaction between the plasma induced by laser radiation and consecutive pulses in the pulse train. Induced electron plasma waves inside the material interfere with incident radiation to produce interference patterns [45,50] that result in permanent periodic modifications. Subsequently, longer periods that result from higher wavelengths enable easier liquid penetration into the nanogratings due to the capillary forces and surface-wetting properties [27]. Etching rates that were only slightly lower than the maximum results were achieved by using 2nd harmonic 500 nJ pulse energy or 28.2 J/cm^2^ energy density radiation. Meanwhile, modifications created by the 3rd harmonic radiation show only a few μm/h etching rates. To fabricate 3D structures, the etching rate of modified versus unmodified regions also matters. A high ratio between modified and unmodified material etching rates is required to obtain a high aspect ratio of fabricated structure features. The unmodified material etching rate was evaluated and, subsequently, the selectivity value could be calculated. For some tested etchants, the unmodified material etching rate value was in the range of the accuracy of the used device. Therefore, only approximate selectivity values are provided. The results of the unmodified material etching rate and the evaluation of selectivity are shown in Figure 2b. Only the H_2_SO_4_ and H_3_PO_4_ mixture showed a significant unmodified material etching rate of around 3 μm/h. Meanwhile, unmodified material could be barely etched by using NaOH, KOH, or HF. When we try to evaluate selectivity with such a low unmodified material etching rate for NaOH, KOH, and HF, it leads to high selectivity values starting from 1000. The high etching rate of unmodified material with sulphuric and phosphoric acid leads to a low selectivity value of 66; despite that, this etchant shows the highest etching rate of modified material.

In this work, our main goal is to demonstrate 3D structures that come out of sapphire. Thus, after single lines tests, 3D tests were performed to find out the protocol that could allow us to obtain 3D structures. Identical parameters set for the single lines test were investigated. Additionally, various pulse durations were tested. Cylindrical structures through the whole sample depth were written into a 500 μm thickness sapphire plate by changing parameters for each cylinder. The results of this experiment can be seen in Figure 3. The 3D experiment has shown that the best parameters, which allow the most efficient etching of the structures, are 1st harmonic radiation with 200 fs duration pulses and 300–500 nJ pulse energies that correspond to 16.9–28.2 J/cm^2^ energy densities of one pulse. The only etchant which etched the structures out of the plate was a H_2_SO_4_ and H_3_PO_4_ mixture. By using other tested etchants KOH, NaOH and HF, cylindrical structures could not be etched out of the plate even after 48 h of etching. This result raised the question of what is the reason the other tested etchants cannot etch out 3D structures.

To test this phenomenon, it was decided to investigate etched and unetched surface modifications and indicate its behavior before and after the etching. Surface modifications were written within the parameter set determined as the most efficient in the previous experiment (1030 nm wavelength radiation, 600 kHz pulse repetition rate, 200 fs pulse duration, and 500 nJ pulse energy). Written 10 μm deep and 10 μm wide modifications on the surface of the sample were etched in different etchants. Scanning electron microscope (SEM) pictures of variously etched and unetched modifications are presented in Figure 4. Moreover, the depth of the etched modifications were measured in each case. In Figure 4a, an unetched modification can be seen right after the laser processing. Before the etching, material modifications are covered by residue remaining after the processing. The unetched modification already has a depth of approximately 0.5 μm. After 1 h of etching in 35% of KOH and 25% of NaOH at 90 °C and at 48% of HF, all the residue is etched and nanogratings are uncovered. Pictures of nanogratings after the etching are depicted in Figure 4b–d. Depths of modifications etched in KOH, NaOH and HF are 3 μm, 3 μm and 7 μm, respectively. In contrast, after the etching modification only for 20 min in the H_2_SO_4_ and H_3_PO_4_ mixture, not only is there residue of lasering, but the nanogratings themselves are etched, which can be seen in Figure 4e. The depth of the etched groove is 9 μm. All these etchants react to material differently. Visually, HF etches out impurities from the surface the most efficiently, meanwhile, after etching in NaOH, the highest amount of residue remains. In comparison with single lines results, HF tends to etch everything from the surface, NaOH and KOH tends to penetrate inside nanogratings. Nevertheless, none of these etchants does etch nanogratings, or in other words, walls between laser-created pores. Thus, KOH, NaOH, and HF etchants cannot etch out 3D structures because part of the laser affected material remains unetched. This insight provokes us to think that not all material in nanogratings becomes amorphous or the amorphized material partly recrystallizes after laser-material interaction stops. Due to the crystalline phase dominant in the nanogratings, HF, KOH and NaOH cannot be used effectively because it does not etch crystalline sapphire, as shown in Figure 2. On the other hand, H_2_SO_4_ and H_3_PO_4_ could etch both crystalline and amorphous materials.

After all the previous insights were made, the most efficient tested protocol was used for structure fabrication out of crystalline sapphire. A few basic 3D structures were fabricated for demonstration. These structures were fabricated out of a 0.5 mm C-cut sapphire plate using H_2_SO_4_ and H_3_PO_4_ mixtures as an enchant. The first demonstrational structure was a hole array. In one case, holes of 100 μm diameter through the complete height of the plate with a pitch of 1 mm were produced, see Figure 5a,b. Additionally, a narrow single Z scanning line channels fabrication with a pitch of 10 μm is demonstrated in Figure 5c,d. In Figure 5, it can be seen that, after the etching, the etched walls are not perfectly vertical. The edges of the structure have non-written planes cut and it forms after the etching due to the sapphire crystalline lattice. This is caused by the high etching rate of unmodified crystalline material. Subsequently, the honeycomb structure was produced out of sapphire. A structure with a 400 μm wall of hexagon and a 200 μm wall thickness between hexagons is shown in Figure 5e–g. The surface roughness of the mentioned structure was measured using an optical profilometer by placing the structure on its side giving a value of 300 nm RMS. This value is comparable with those shown in fused silica SLE-made surfaces [51]. The lowest demonstrated etched fused silica surface roughness is approximately 100 nm RMS. Surface roughness is mostly determined by the morphology of induced nanogratings and how effectively material between nanogratings could be etched. Since fused silica is a less chemically inert material than sapphire, the expected etched sapphire structure roughness should be higher than for the same surfaces made in fused silica. To show the versatility of this technology, similar honeycomb designs with 100 μm and 50 μm wall thicknesses were also made (Figure 5h,i). Walls thinner than 50 μm were fragile and did not survive the etching procedure. Since we were working with crystalline sapphire, it shows different etching rates of unmodified material in various crystalline orientations [43]. Thus, the tendency of a repeating sapphire crystalline structure can be seen in honeycomb structures. The edges of etched structures appear to have a chamfer and are inclined according to a sapphire crystalline lattice. Nonetheless, these structures show the possibility of producing 3D structures out of crystalline sapphire. Due to the high mechanical and chemical resistance of sapphire, such structures show potential as micromechanical components in complex miniaturized systems.

## 4. Discussion

In SLE, we always desire to have high selectivity and a high etching rate. High selectivity values are needed to maintain good accuracy and a high aspect ratio of etched features. On the other hand, a high etching rate is required for fast processes. The main problem is that, usually, these two are achieved with different etching protocols. This means that, at the same time, we cannot obtain both, but in one structure, it is possible to combine both etching protocols. For fused silica, this was already demonstrated [52] when structure parts that do not require precision are etched with HF and precise features are etched with KOH solutions which give the highest selectivity. The same could be done with the etching of sapphire by combining KOH, NaOH, or HF etching with a sulphuric and phosphoric acid mixture. In that way, etching time with H_2_SO_4_ and H_3_PO_4_ can be minimized and, as a result, higher accuracy could be obtained.

## 5. Conclusions

In this work, we have presented a comparison between different sapphire SLE processing algorithms. During previously shown tests, various laser parameters for creating laser-irradiated volume were tested. The best results were obtained with 1030 nm wavelength 200 fs pulse duration radiation, which could be explained by higher periodicity nanogratings and easier etchant penetration. Moreover, four different etchants—HF, NaOH, KOH, and H_2_SO_4_ with H_3_PO_4_—were tested. The only etchant which allows the total removal of laser-modified sapphire is a H_2_SO_4_ with H_3_PO_4_ mixture, while the other tested etchants only etch material in between laser-formed nanogratings. After a moderate protocol, a few structures were found: hole matrix and honeycomb structures were fabricated out of C-cut crystalline sapphire. We noticed that sapphire repeats its crystalline lattice during the etching due to different etching rates in separate crystalline orientations. This property leads to less accurate features of the fabricated object and self-organized patterns on the structure’s surface. However, this effect could be diminished by combining two different etching steps for the same structure. Thus, in this work, we have demonstrated a protocol that allows 3D structure formation out of crystalline sapphire. However, additional improvements, such as etching protocol optimization or combined etching development, could be made to the protocol to refine the quality of the produced structures. These results will contribute to the rapid SLE fabrication of future devices.

## Figures and Tables

**Figure 1 micromachines-14-00007-f001:**
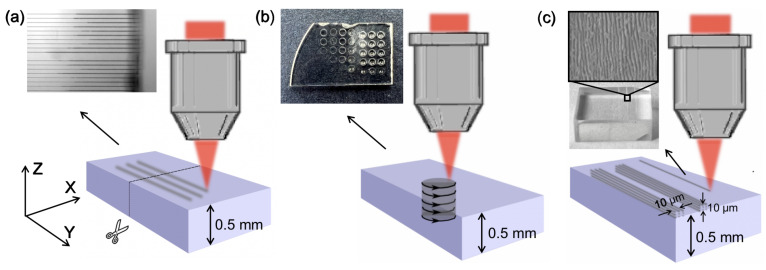
The schemes and the resulting pictures after the etching of performed experiments. (**a**) Single lines experiment when lines from a single scanning are written in the volume of material to test the etching rate of modified material. (**b**) Scheme of 3D experiments when cylindrical structures are written through the whole plate thickness. (**c**) Laser-induced surface changes for modification morphology observation after the etching procedure.

**Figure 2 micromachines-14-00007-f002:**
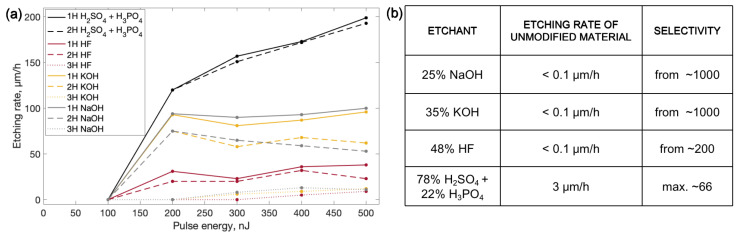
(**a**) Graphs of etchings rates of single lines written and etched by different processing protocols. 1H, 2H and 3H stand for 1st, 2nd and 3rd harmonics of 1030 nm wavelength laser, which is 1030 nm, 515 nm and 343 nm, respectively. (**b**) Table of the etching rate of unmodified material etched by different etchants and selectivity evaluation according to modified and unmodified material etching rate results.

**Figure 3 micromachines-14-00007-f003:**
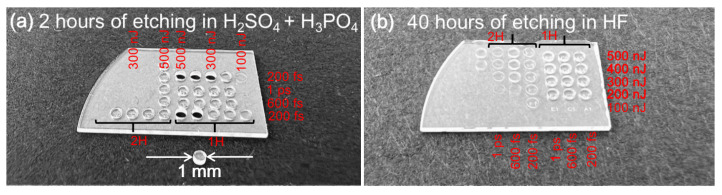
Cylindrical structures written with different wavelengths and pulse duration radiation. (**a**) Structure etched in H_2_SO_4_ and H_3_PO_4_ mixture, (**b**) structure etched in HF acid.

**Figure 4 micromachines-14-00007-f004:**
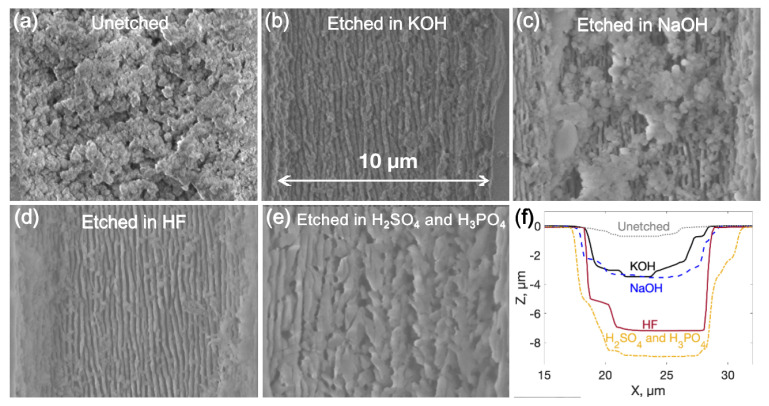
SEM pictures of laser-irradiated surface morphologies and measured profiles of the processed material. (**a**) Unetched surface modification, (**b**) surface modification etched in 35% KOH solution for 1 h, (**c**) surface modification etched in 25% NaOH solution for 1 h, (**d**) surface modification etched in 48% HF for 1 h, and (**e**) surface modification etched in H_2_SO_4_ and H_3_PO_4_ for 20 min. (**f**) Optical profilometer measured processed profiles of all presented modifications.

**Figure 5 micromachines-14-00007-f005:**
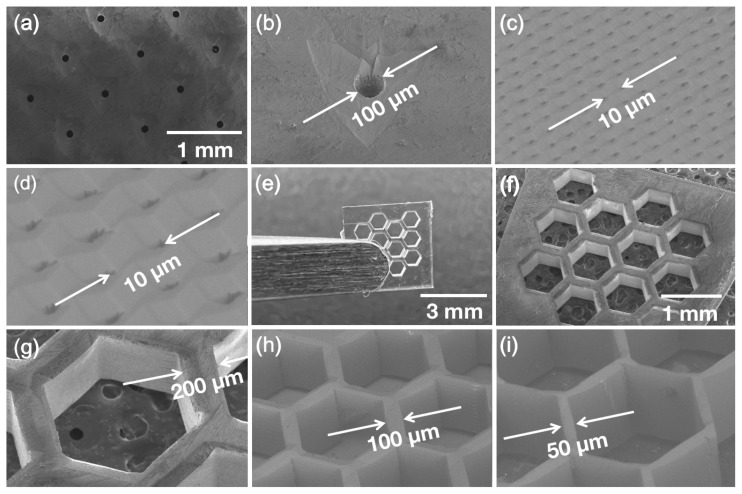
Structures fabricated out of crystalline C-cut sapphire. (**a**,**b**) 100 μm diameter hole matrix, (**c**,**d**) single line 10 μm pitch hole matrix, (**e**–**g**) honeycomb structure where the side of the hexagon is 400 μm, and the walls between hexagon are 200 μm, (**h**,**i**) similar hexagon structure with diminished walls between hexagon respectively 100 μm and 50 μm.

## Data Availability

Not applicable.

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
