# Peer review of "Sapphire Selective Laser Etching Dependence on Radiation Wavelength and Etchant"

_micromachines, 2022, doi:10.3390/mi14010007_

Round 1
Reviewer 1 Report (New Reviewer)
This article discusses the results of the selective etching of sapphire modified by femtosecond lasers with different etchants, and experimentally obtains the best parameters for each etchant to be used in the experiments, and explains why some etchants cannot be used for 3D engraving, and finally uses the selected etchant in combination with laser modification to prepare honeycomb structures on sapphire. However, the logic, the English presentation and the image processing of the paper still need further improvement.
1.The authors explain the reasons why he choose the different reaction temperatures for different etchants.
2. The author explains the reasons for the different etching effects of different etchants. Different etchants have different corrosion effects on the surface residues. Only the mixture of phosphoric acid and sulfuric acid can quickly etch the sample,and the others can only remove the residue.
3. There are many literatures on the on the ultrafast laser-material interactions. The author should compare its advantages and disadvantages with the literature, such as:
[1] Y. Song, C. Wang, X. Dong, K. Yin, F. Zhang, Z. Xie, D. Chu, J. Duan, Controllable superhydrophobic aluminum surfaces with tunable adhesion fabricated by femtosecond laser, Opt. Laser Technol. 102 (2018) 25-31.
[2] Jia X.S., Chen Y.Q., er al, Combined pulse laser: Reliable tool for high-quality, high-efficiency material processing, Opt. Laser Technol., 2022, 153:108209.
[3] K. Ding, C. Wang, S. Li, X. Zhang, N. Lin, J. Duan, Large-area cactus-like micro-/nanostructures with anti-reflection and superhydrophobicity fabricated by femtosecond laser and thermal treatment, Surf. Interfaces 33 (2022) 102292.
[4] Ding K.W., Wang C., Li S.H., Zhang X.F., Lin N. and Duan J.A., Single-step femtosecond laser structuring of multifunctional colorful metal surface and its origin, Surf. Interfaces, 2022, 34:102386.
4. There are still some grammatical errors in the article as follows:
[1] In line 51, suggest should be changed to suggests to ensure subject-verb agreement
[2] In line 103, "ambient" should be preceded by the definite article “the”.
[3] In line 128, “hour” should be changed to “hours”.
[4] In line 191, "unetched" should be preceded by the definite article “the”.
[5] In line 200, remain should be changed to remains to ensure subject-verb agreement
[6] In line 228,the “Thinner walls than 50 µm” should better changed to “wall thinner than 50µm”
[7] In line 231, "tedency" should be preceded by the definite article “the”.
[8] In line235, the article “a” should be removed.
Author Response
Dear Reviewer,
thank you for all the comments. Responses are provided in the attached file.

Reviewer 2 Report (New Reviewer)
The authors demonstrate a comparison between different wavelength radiation usage for modification inscription and various etchants comparison. A few structures: hole matrix and honeycomb structures, were fabricated out of crystalline sapphire. The manuscript can be published after revisions. However, the authors should consider the following comments/questions:
1. In 1D experiments, why the glass plate is selected as the research object instead of sapphire?
2. In Fig.3, I suggest to label the experimental parameters, such as pulse durations and energies.
3. The influence of wavelength on etching results is mentioned and the optimum parameters are selected. It can be considered to adding the comparison of etching results for different wavelength parameters.
4. Etching of nano-grating structures is studied in this paper, but there is no picture of nano-gratings. It can be considered to enlarge the result pictures of Figure 1 or show the result picture before etching in Figure 3.
Author Response
Dear Reviewer,
thank you for all the comments. Responses are provided in the attached file.

Reviewer 3 Report (New Reviewer)
See attachment for my feedback

Author Response
Dear Reviewer,
thank you for all the comments. Responses are provided in the attached file.

This manuscript is a resubmission of an earlier submission. The following is a list of the peer review reports and author responses from that submission.
Round 1
Reviewer 1 Report
This article relates to the microstructuring of transparent solid materials using laser fs exposure with further etching. It presents several algorithms for SLE processing of sapphire. Several wavelengths and several etchants have been investigated. The authors obtained the best results with radiation at a wavelength of 1030 nm, a pulse duration of 200 fs, and using a mixture of H2SO4 with H3PO4 as an etchant.
At the same time, the presented article has significant and minor shortcomings.
Significant disadvantages include:
1. The presented protocol with a mixture of sulphuric and phosphoric acids cannot be considered optimal. The fact is that the results are presented for a random ratio of components (78%/22%) and a random temperature of 270 °C. Such a ratio should be substantiated either from literary sources or from experiment. As for the temperature of the mixture, it is chosen quite high. Its value must also be substantiated, for example, experimentally by comparing the results for several values of the parameter. If we talk about the Arenius temperature acceleration of the etching rate, then it is also necessary to justify why such high temperatures were not applied to other etchants used in the article. At the same time, KOH and NaOH solutions were heated to 90 °C, and the effectiveness of HF, for some unknown reason, was tested only at 20 °C. To work out the technological process, all the presented etchants must be tested in the same temperature range and the results are summarized in a single table.
2. When choosing the optimal wavelength, the authors performed experiments using the same waist radius for all wavelengths. In this case, the energy characteristics used in the laser spot are indicated only for the first harmonic. The text must be supplemented with a table indicating for all wavelengths such energy characteristics as power and maximum fluence for one pulse, as well as frequencies. Due to the lack of this information, it is impossible to understand why different efficiencies are obtained for different wavelengths.
Minor comments include:
1. Line 48
The authors write: “More than a decade ago, the first ideas of SLE processed in sapphire were published [30]. This publication suggest inscribing nanogratings inside the sapphire volume and etching it with KOH.". These proposals must be preceded by a text stating that the use of KOH to accelerate laser etching of sapphire was first proposed in the article [Dolgaev SI, Karasev ME, Kulevskii LA, Simakin AV, Shafeev GA. Dissolution in a supercritical liquid as a mechanism of laser ablation of sapphire. Quantum Electronics. 2001 Jul 31;31(7):593].
2. The Discussion section should be substantially expanded. Here, with appropriate references, it is necessary to describe in detail the features of the results obtained and explain them from the physical and chemical points of view.
It is also desirable to correct less significant shortcomings:
1. For better perception of the text, all large paragraphs (beginning on lines 58, 110, 146, 169) should be divided into several.
2. Line 30
move here the description of the abbreviation "Selective laser etching (SLE)" from line 32.
3. Line 66
The phrase "Radiation wavelength itself could change the size and periodicity of nanaogratings [33] which later affects the etching properties of inscribed modifications." need to be rewritten and expanded to make the authors' thoughts more understandable to the reader.
4. Line 86
"The light was linearly polarised perpendicularly to the scanning direction to obtain etching rates as high as possible". Explain physically with an appropriate reference why the maximum etching rate is achieved with this polarization.
5. Line 108
In Figure 1c, it is not clear what the arrows with dimensions of 10 µm refer to? Show more clearly.
6. Line 114
Indicate whether this is the average or maximum energy in the laser spot.
After making the appropriate changes and additions, the article can be published.
Reviewer 2 Report
The authors report their research on sapphire microprocessing by combining femtosecond laser with various etchants for selective laser etching (SLE). In their work, they examined the results of SLE by properly optimizing the laser parameters and varying the etchants. I agree with the experimental data. However, I can‘t find the significance and novelty that is enough for publication in Materials. The authors merely present the experimental results, lacking in in-depth discussion. On the other hand, the manuscript is not well written, because there is still much room for improvement in language. The following just shows several representative writing errors:
1) There have been already shown various laser machining techniques…laser-induced plasma-assisted ablation [4,5] or back side wet etching [6–8].
2) Only by optimizing laser parameters selectivity above 1000 could be obtained for fused silica [15].
3) Optimizing the process with particular burst regimes [19] or adding organics to the etchant [20] selectivity above 2000 could be obtained.
4) The etching selectivity values of around 10000 [32] obtained.
Reviewer 3 Report
Presented manuscript shows some applicable results in the important area of precise processing the transparent materials. Although overall research background seems to be of high importance, there are certain concerns regarding article under consideration:
1. Authors claim in the Intro section (rows 53-54) that "no one has ever demonstrated SLE-made 2D or 3D objects out of sapphire". This bold statement contradicts with the results provided by Gottmann 2009 and Capuanoa 2020, cited in the article as sources [33] and [35] respectively. Moreover, similar and sometimes more impressive results were already shown by Wortmann 2008, Hörstmann-Jungemann 2009 & 2010, Moser 2011, Xie 2019, Kaiser 2019, Capuanoa 2022, Liu 2022, and Gottumukkala 2022, to name a few. All these researches and some of the others were unfortunately omitted from the introduction provided, and need to be thoroughly researched and cited properly to give a context for the results presented.
2. To continue with suggestion #1, there seems to be a lack of comparison between the new achieved results and the results shown in the already published works. The quantative analysis of the achieved etching rates, profile depths, and structures um scales needs to be provided in comparison with other works.
3. The declared dependence on radiation wavelength is not clearly shown in the Results section, and the "best results were obtained with 1030 nm wavelength 210 200 fs pulse duration radiation" claim in the Conclusions (rows 210-211) stays unsupported. There more thourough discussion needs to be provided, with the analysis of the physical mechanisms of the influence (presumably due to difference in absorption, but it needs to be checked)
4. It is hardly believeable that the 1 mm-wide honeycomb structures shown at fig5e-i are related to the internal crystalline structrure of the material rather than to the scanning trajectory shown at fig1b, so the "sapphire repeats its crystalline lattice during the etching" bit in the Conclusions (rows 216-217) stays unsupported.
5. What are the possible application areas for the 100 um, and even more so, for the 1 mm wide structrures?
6. What are the physico-chemical mechanisms behind the arranged submicron structures ("nanogratings"), clearly visible at fig 4b & 4d (after etching specifically in KOH and HF), and not seen at fig 4a, 4c and 4e? The explanation of this difference might be helpful.
Reviewer 4 Report
The manuscript reported using the SLE method to process the 3D micro-nano structure on the sapphire. The influences of different femtosecond laser wavelengths and different etching agents on processing sapphire were explored. Through the control experiments, the solution consisting of H3PO4 (the mass fraction of 78%) and H2SO4 (the mass fraction of 22%) with a temperature of 270 °C to was able to fabricate the micro-nano structures on the sapphire at a high etching rate. However, I have some comments as follows.
(1)There are four types of etching conditions to process the sapphire in the experiment: the KOH solution (35% mass fraction, 90 °C), the NaOH solution (25% mass fraction, 90 °C), the HF solution (48% mass fraction, 20 °C), and the mixture solution consisting of H3PO4 (the mass fraction of 78%) and H2SO4 (the mass fraction of 22%) at 270 °C. And the authors found that the mixture solution could promote the etching rate of the sapphire. However, for the KOH solution, NaOH solution, and HF solution, whether the parameters of temperature and mass fraction are the best? Are the results with these etchants comparable to that of using the mixture solution?
(2)Some recently published references about etching on semiconductor are suggested to added into the introduction.International Journal of Extreme Manufacturing, 2021, 3(3): 35104; Small Methods, 2022, 6:2200329
(3) How is the etching rate measured in this manuscript? Please describe clearly the measurement method.
(4) The illustrations in the manuscript are not clear, such as Figure 1. And a partially enlarged view should be added in Figure 4 so as to make the manuscript readable.
(5) In line 143 of this manuscript, if the KOH, NaOH and HF solutions of 270°C were used as the etching agent respectively, will the processed 2D microstructure peel off the substrate?
(6) The authors are suggested to clearly describe how to safely heat the mixture of H3PO4 and H2SO4 to 270 °C as it will help the readers to repeat the experiments.